# AMG-Embedding: a Self-Supervised Embedding Approach for Audio Identification

Yuhang Su
Beijing University of
Chemical Technology
Beijing, China
2022200801@mail.buct.edu.cn

Wei Hu*
Beijing University of
Chemical Technology
Beijing, China
huwei@mail.buct.edu.cn

Fan Zhang
Beijing University of
Chemical Technology
Beijing, China
zhangf@mail.buct.edu.cn

Qiming Xu
Beijing University of
Chemical Technology
Beijing, China
2022200846@buct.edu.cn

## Abstract

Audio Identification aims to precisely retrieve exact matches from a vast music repository through a query audio snippet. The need for specificity and granularity has traditionally led to representing music audio using numerous **short fixed-duration overlapped segment/shingle** features in *fingerprinting* approaches. However, *fingerprinting* imposes constraints on scalability and efficiency, as hundreds or even thousands of embeddings are generated to represent a music audio. In this paper, we present an innovative self-supervised approach called Angular Margin Guided Embedding (AMG-Embedding). AMG-Embedding is built on a *traditional fingerprinting* encoder and aims to represent **variable-duration non-overlapped segments** as embeddings through a two-stage embedding and class-level learning process. AMG-Embedding significantly reduces the number of generated embeddings while achieving high-specific fragment-level audio identification simultaneously. Experimental results demonstrate that AMG-Embedding achieves retrieval accuracy comparable to the *based fingerprinting* approach while consuming less than $1/10th$ of its storage and retrieval time. The efficiency gains of our approach position it as a promising solution for scalable and efficient audio identification systems.

## CCS Concepts

• **Information systems → Information retrieval**; • **Computing methodologies → Artificial intelligence**.

## Keywords

audio identification, fingerprinting, music embedding, class-level learning

**ACM Reference Format:**
Yuhang Su, Wei Hu, Fan Zhang, and Qiming Xu. 2024. AMG-Embedding: a Self-Supervised Embedding Approach for Audio Identification. In *Proceedings of the 32nd ACM International Conference on Multimedia (MM '24), October 28-November 1, 2024, Melbourne, VIC, Australia.* ACM, New York, NY, USA, 10 pages. https://doi.org/10.1145/3664647.3681647

*Corresponding author

## 1 Introduction

Audio identification, often referred to as audio fingerprinting[3], is an audio-based music information retrieval (MIR)[17] task. It plays a crucial role in various applications, including the identification of unknown music from microphone input, the detection of duplicated music tracks. Precisely recognizing music from a vast database when presented with an input audio snippet, especially when the snippet is just a small fragment exposed to signal distortions like noise, reverberation, or compression, poses a significant challenge.

Audio identification is known for its exacting demand in terms of specificity and granularity, involving *high-specific fragment-level* retrieval tasks[17]. This implies that an audio identification system is expected to precisely return copies of the query, which might consist of only a short fragment. To meet the high-specificity fragment-level requirement, cutting-edge audio identification approaches [4, 16, 35, 60] opt to divide music audios into numerous consecutive short fixed-duration overlapped segments or shingles. Each segment is represented as an $L^2$-normalized feature/embedding, where the inner product of features can measure similarities between segments. Therefore, audio identification can be viewed as a feature sequence matching problem. Moreover, owing to the fragment-level demands of audio identification, the duration and overlapping length of these segments are typically quite short. For instance, in *neural fingerprinting*[4], the duration is 1$s$ with a 0.5$s$ overlap. As a result, a typical music track, even one lasting several minutes, may necessitate hundreds of such features for representation. Consequently, these audio identification approaches face scalability and efficiency challenges due to the expanding feature repositories in large-scale music databases.

Recently, representing a long variable-length music with an embedding, known as Music Embedding, has demonstrated success in MIR, such as music classification[62], recommendation [23], and version identification[14]. However, existing Music Embedding methods primarily target *low-specificity document-level* MIR tasks and may not simultaneously meet the specificity and granularity requirements of audio identification.

In this paper, we introduce a novel self-supervised Angular Margin Guided Embedding approach (AMG-Embedding) for music audio identification[1]. AMG-Embedding aims to represent variable-duration (but with an upper limit, for example, less than 30 seconds) audio segments as a single embedding, while ensuring that the inner product can still measure the similarity between segments with different durations. Moreover, there is no longer a necessity for overlap between segments. Hence, AMG-Embedding allows us to represent music with relatively few segments and embeddings.

[1]Code is available at https://github.com/yuhangsu82/AMG-Embedding.

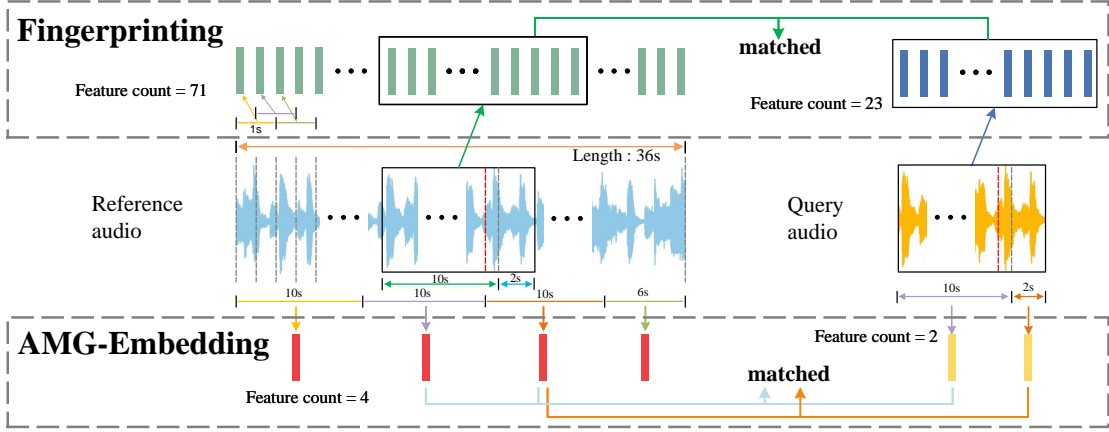

**Figure 1: Audio identification using traditional fingerprinting and our AMG-Embedding. To measure similarity between a** $36s$ **audio and a** $12s$ **query audio which is actually a segment of the former, the fingerprinting method requires generating 94 features in total, whereas AMG-Embedding only requires 6 features.**

As illustrated in Figure 1, consider a $36s$ audio, 71 features are required to represent the entire audio by using the traditional fingerprinting methods with fixed-duration segments of $1s$ and $0.5s$ overlap. In contrast, using AMG-Embedding, we can segment the audio into 4 non-overlapping segments (three $10s$ and one $6s$) with an upper duration limit of $10s$($Dur_{max}$ as referred to in the paper), requiring only 4 embeddings to represent the entire audio. Therefore, in the case of the $12s$ querying audio, the fingerprinting generates 23 features, while AMG-Embedding only produces 2 embeddings. Furthermore, **inner product of features can still measure the similarity between corresponding segments, even with different durations**. In other words, the more identical content shared between two audios, the larger the inner product of their features. Consequently, both long music pieces and short snippets can be represented with much shorter feature sequences compared to existing fingerprinting systems, allowing feature sequence matching to remain effective for measuring audio similarity.

AMG-Embedding is a two-stage embedding approach. The first stage employs existing fingerprinting approaches to extract feature sequences from input audios, and then these feature sequences are further compressed in the second stage by using a standard Transformer Encoder trained with our proposed class-based Proxy-anchor Aligned Margin loss function PAM-Loss. AMG-Embedding offers a significant advantage in terms of retrieval efficiency and scalability compared to previous fingerprinting methods, while maintaining a comparable level of accuracy. The main highlights of the work are summarized as follows:

(1) We proposed a two-stage approach AMG-Embedding to represent a variable-duration audio as an embedding, which can significantly reduce the size of feature sequences while concurrently meeting the granularity and specificity requirement of audio identification. Moreover, any SOTA fingerprinting approach can seamlessly integrated in AMG-Embedding.

(2) We examined the constraints associated with commonly used proxy-based losses in the context of variable duration audio embedding. In response, we introduced a new proxy-based loss, PAM-Loss, specifically crafted to enhance the efficacy of our embeddings. PAM-Loss also stands as a commendable endeavor to harness class-level learning, aiming to elevate the performance of Music Information Retrieval (MIR) tasks.

(3) We performed experimental analyses, presenting both quantitative and qualitative results to convincingly showcase the robust effectiveness of AMG-Embedding.

## 2 Related Work

### 2.1 Audio Identification

Audio identification is typically tackled in two steps: the extraction of feature/fingerprint sequences from audio, and the construction of indexes for retrieval. Handcrafted features have been extensively employed. Shazam[41] extracts sets of spectral peaks in the spectrogram as fingerprintings. Similar quad combination was adopted in [37]. Binary fingerprints based on energy changes across spectral-temporal space are employed in [19]. Waveprint[2] computes fingerprints using wavelets, and [34] generates fingerprints by using pseudo-sinusoidal components. Nevertheless, handcrafted features often fall short in real-world applications. Recently, Deep learning, particularly unsupervised learning, has been applied and achieves impressive performance in audio identification. Autoencoders consisting of LSTM layers are proposed in [1, 50], and CNN models [4, 16, 35, 57, 60] are widely employed as encoders recently. These encoders are trained to generate embeddings of *each audio segment* using the metric learning loss functions, such as Contrastive loss[26], Triplet loss[16, 50, 60], N-Pairs/InfoNCE/NT-Xent loss[4, 35, 57]. In these approaches, each source music and query snippet is represented as a sequence/set of features for retrieval, which results in the creation of massive feature databases. For example, 56M segment features are generated for a subset of the fma-full[7] consisting only 278s audio clips of 100k songs. To ensure efficient retrieval, audio identification systems must employ fast and effective indexing algorithms, such as hashing[2, 35, 41, 60] and inverted indexing[4, 16].

## 2.2 Music Embedding in MIR

A logical approach is to use a compact embedding to represent either a short audio snippet or a complete music as variable duration embeddings. Embedding-based approaches can obtain compact representations that expedite the retrieval process, reduce storage requirements, and enable efficient similarity estimation. As mentioned above, music embedding have recently found applications in certain *low-specificity document-level* MIR tasks, such as music classification[9, 10, 52, 62], recommendation [5, 23, 42], *version identification*(also referred as *cover song detection, CSI*)[11, 13, 15, 33, 53, 54, 58, 59], etc. In many of them, MIR tasks are formulated as classification or metric learning challenges employing CNN/Transformer-based encoders to map music to their embedding vectors. For instance, Swin Transformer[25] and InfoNCE loss are used in S3T[62] for music classification, ResNet-IBN and classification-triplet joint losses are employed in ByteCover series[13–15] for cover song identification. It should be noticed that ByteCover3 implements short-query CSI by borrowing finger printing techniques in audio identification, and its segment length is **fixed** 20*s* with 10*s* overlapping. In addition, several self-supervised learning approaches [31, 38, 55, 63] have emerged with a focus on learning music representation (music embedding). These embeddings can be applied in various downstream music-related tasks, also primarily in music classification and *version identification*. Embedding solutions have not yet been effectively applied to audio identification due to their limitations in terms of specificity and granularity.

## 2.3 Metric Learning

Metric learning seeks to map data into an embedding space where similar data points cluster together, while dissimilar ones are separated by a significant margin. There are two main paradigms in metric learning: **class-level** learning and **pair-wise** learning.

Class-level learning usually incorporates a weight matrix, where each column (serve as a **proxy**) corresponds to a particular class in order to transform the embedding space into class probability vectors. These methods are also known as **proxy-based** learning. The most basic approach is the normalized softmax loss[44, 61], where the columns of the weight matrix are L2 normalized. A variation of this approach is ProxyNCA[27], which employs cross entropy on the Euclidean distances between embeddings and the weight matrix. Several additive angular margin loss functions, including SphereFace[24], SphereFace2[49], CosFace[43, 45], and ArcFace[8], further modify the cross entropy loss with angular margins, and achieve great improvement. Additionally, the SoftTriple loss[30] expands the weight matrix to have multiple columns per class to provide more flexibility.

Pair-wise learning directly learns similarity between training samples in the embedding space and does not require proxies. Contrastive loss[18] and triplet loss[47] are two fundamental approaches. Numerous losses have been developed based on them, including the angular loss[46], the margin loss[51], the lifted structure loss[28]. Recently, mining more relationships (positive and negative pairs) of samples becomes popular, such as the N-Pair loss(also known as InfoNCE and NT-Xent)[6, 29, 36], the tuplet margin loss[56], and the PSL[48].

Metric learning is widely adopted for addressing MIR tasks. However, it's important to note that despite its notable success in some retrieval approaches, there has been relatively limited emphasis on class-level learning in the context of MIR. In many scenarios, pair-wise learning is preferred because it is more convenient to specify similarities between samples using pair or triplet relationships. Furthermore, it's essential to highlight that **current class-level/proxy-based learning approaches are not particularly suitable for variable duration audio embeddings**. A more detailed discussion on this topic will be provided in Section 3.3.1.

## 3 Proposed Approach

A self-supervised embedding-based audio identification approach, named AMG-Embedding, is proposed in this paper. Embedding-based methods are well-suited for low-specificity document-level MIR tasks. However, their application in audio identification has proven to be elusive. To tackle this issue, AMG-Embedding employs a **two-stage embedding** strategy tailored for audio identification.

### 3.1 Two-Stage Embedding

AMG-Embedding is a two-stage embedding approach, where a piece of variable-duration music audio is converted into a sequence of segment features by using a self-supervised fingerprinting system (*neural fingerprinting*[4] in our implementation) in the first stage (named **Fingerprinting Stage**), then this sequence is further compressed to a much shorter embedding sequence by employing a transformer-based[35] encoder trained with PAM-Loss, a proxy-based metric loss, in the second stage (named **Embedding Stage**).

**Fingerprinting Stage**. In this stage, we adopt a self-supervised *neural fingerprinting* approach[4]. As illustrated in Figure 2, initially, we segment an input audio $X$ into individual segments, denoted as

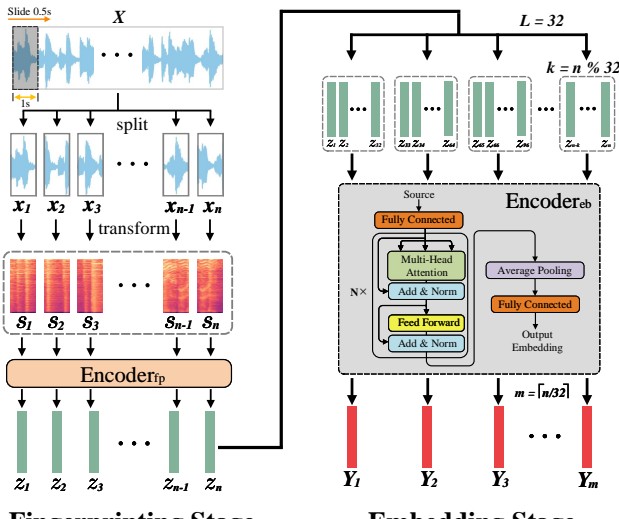

**Figure 2: An audio $X$ is transformed to a short sequence of features $\{Y_i\}_{i=1}^{\lceil n/L \rceil}$ with our Two-Stage Embedding. AMG-Embedding has the potential to reduce the feature quantity to $1/L$ of the based fingerprinting method.**

$\{x_t\}_{t=1}^n$, achieved through time slicing. Each $x_t$ represents a segment unit of $X$ at time step $t$. Subsequently, we transform $\{x_t\}_{t=1}^n$ into the corresponding time-frequency representation, denoted as $\{s_t\}_{t=1}^n$. The number of $s_t$ mirrors the duration of the input audio $X$. Subsequently, we utilize a fingerprinting encoder referred to as $Encoder_{fp}$, to project $\{s_t\}_{t=1}^n$ into a sequence of $L^2$-normalized $d$-dimensional embedding vectors $\{z_t \in \mathbb{R}^d\}_{t=1}^n$ used for retrieval in common fingerprinting systems. Due to the need for granularity at the fragment-level, each segment must have a small time duration, and adjacent segments also need to have an overlap, for instance, the segment length is 1s and the overlap is 0.5s indicated in [4].

**Embedding Stage**. In this stage, the sequence $\{z_t\}_{t=1}^n$ should be further compressed into a new much shorter feature sequence using $Encoder_{eb}$. In our implementation, $Encoder_{eb}$ is a standard **Transformer Encoder**[40] in which an *Average Pooling* layer and a *Fully Connected* layer are applied to output the final embedding. It's worth noting that $Encoder_{eb}$ comes with **an input size limitation** of $L$ (corresponding to the upper duration $Dur_{max}$), which results in the sequence $\{z_t\}_{t=1}^n$ being divided into $\lceil n/L \rceil$ *non-overlapping* sub-sequences. Each sub-sequence is then independently fed into $Encoder_{eb}$ to generate its final fixed-length $L^2$-normalized $f$-dimensional embedding $Y_i \in \mathbb{R}^f$. Consequently, a variable-duration audio $X$ is mapped to $\lceil n/L \rceil$ embeddings, denoted as $\{Y_i\}_{i=1}^{\lceil n/L \rceil}$. In our implementation, $L$ is typically set to 32 or 64 (corresponding to about 15$s$ and 30$s$ in duration), significantly reducing the embedding size. **Note** that $f$ can differ from the dimension of the fingerprinting embedding $d$, but in our implementation, we set $f = d$ because the objective of AMG-Embedding is reducing feature. Moreover, the same dimension of $z_i$ and $Y_i$ also can simplify **Proxy Initialization** process introduced in Section 3.3.2.

As the result, a music is represented as a sequence of short embedding vectors after Embedding Stage. Consequently, these embedding vectors can be used to build indexing for searching. $Encoder_{fp}$ and $Encoder_{eb}$ are employed in AMG-Embedding for two-stage embedding as $X \rightarrow \{z_t\}_{t=1}^n \rightarrow \{Y_i\}_{i=1}^{\lceil n/L \rceil}$. The choice to structure AMG-Embedding as a two-stage method can **be attributed to two factors**:

(1) By replacing $Encoder_{fp}$, any state-of-the-art fingerprinting approach can be seamlessly integrated into AMG-Embedding, thereby enhancing the capabilities of AMG-Embedding. This allows for the incorporation of advanced techniques to improve the overall performance of the embedding.

(2) Combining the two stages into a unified process, specifically, training one encoder to replicate the current performance, introduces notable challenges. The requirement for an enlarged model input size brings about a tenfold escalation in model complexity and parameter count. Consequently, the training of the consolidated encoder becomes difficult to achieve performance parity with our two-stage solution.

$Encoder_{eb}$ plays a pivotal role in AMG-Embedding, and it actually compresses embeddings by projecting a variable-length sequence of $k$ features ($1 \le k \le L$) derived from an audio segment into a final embedding $Y$, and the inner product of $Y$ serves as a metric for measuring the similarity between corresponding segments.

The model architecture, the chosen metric loss, and the training paradigm of $Encoder_{eb}$ should be carefully designed.

## 3.2 Training Pipeline of $Encoder_{eb}$

Figure 3 illustrates the training pipeline of $Encoder_{eb}$. AMG-Embedding employs a standard **Transformer Encoder** as $Encoder_{eb}$ to compress feature sequences, and class-level metric learning is employed to train $Encoder_{eb}$.

Firstly, independent audios are collected as training data, with specific segments (*anchor zone* in the figure), whose durations are $Dur_{max}$, chosen as the **anchor samples** for each audio. Each specific anchor is associated with a distinct class in class-level learning. Moreover, we subject these audios to **fragment&quality augmentation** to simulate signal distortions: a segment of the original audio, with a duration also not exceeding $Dur_{max}$, is extracted firstly. Importantly, these segments must exhibit **content overlap** with their corresponding anchor. Then the quality augmentation is applied to these segments to generate the **augmented samples** through the same steps as *neural fingerprinting*[4]: time offset modulation, background mixing, impulse response filtering, cutout, and spec-augment.

After the *Fingerprinting* Stage, both **anchor** and **augmented** samples are converted into feature sequences, and then these sequences are transformed into final embeddings by using $Encoder_{eb}$, and the embeddings of an anchor sample and its augmented counterparts share the same unique class label. The assignment of identical class labels to both the anchor samples and their augmented replicas suggests that the embedding of an audio segment exposed to signal distortions should exhibit greater similarity to the embedding of its exact musical match than to embeddings of other music tracks. **This constitutes a critical aspect of audio identification**, ensuring accurate matching of distorted or fragment-level audios to their original sources.

In theory, any class-level metric loss functions, such as ArcFace[8], can be utilized to train $Encoder_{eb}$ by encouraging each final embedding to approach its class proxy while simultaneously moving away from proxies of other classes. However, we found that common class-level metric losses are not suitable for this task (discussed in Section 3.3.1). Therefore, we proposed Proxy-anchor Aligned Margin Loss (PAM-Loss) to significantly solve the problem.

## 3.3 PAM-Loss

PAM-Loss is a proxy-based loss function, derived from ArcFace-Loss[8], primarily designed to improve variable-length embeddings. Let's first revisit ArcFace-Loss and analyze why it is not an appropriate choice for training $Encoder_{eb}$.

*3.3.1 ArcFace-Loss Revisited.* Consider a training batch comprising $N$ samples, denoted as $\{Y_i, l_i\}_{i=1}^N$, where $Y_i$ is the embedding of a training sample and $l_i$ indicates its class label. ArcFace-Loss defines the angle $\theta_j$ between $Y_i$ and the $L_2$-normalized center (**proxy**) of the $j$-th class, represented as $P_j \in \mathbb{R}^f$, where $\cos \theta_j = P_j^T \cdot Y_i$. The primary objective of ArcFace-Loss can be expressed as

$$L_{arc} = \frac{1}{N} \sum_{i=1}^N (-\log \frac{e^{s \cos(\theta_{l_i}+m)}}{e^{s \cos(\theta_{l_i}+m)} + \sum_{j \neq l_i} e^{s \cos \theta_j}}) \qquad (1)$$

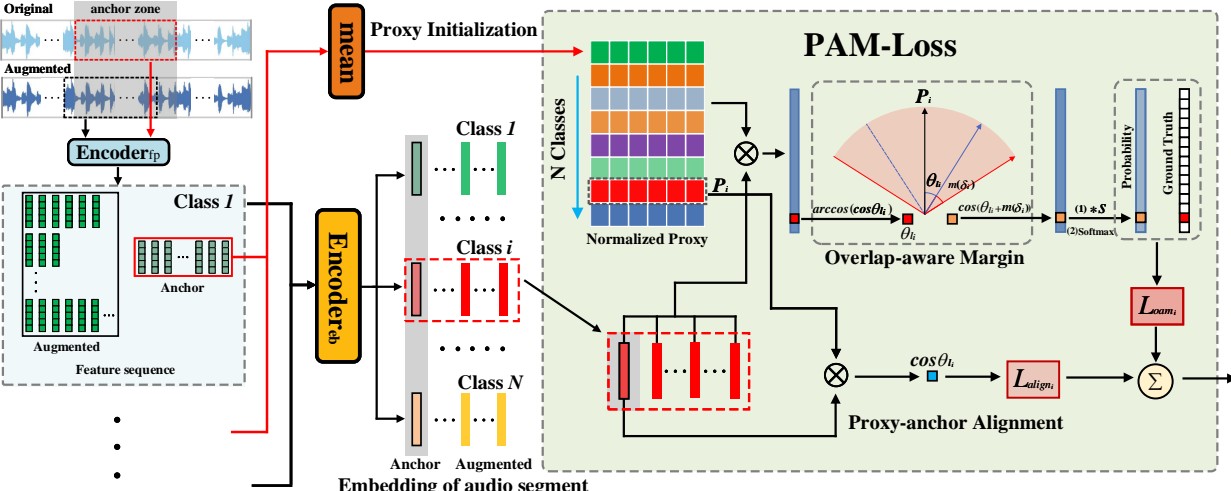

**Figure 3: Training pipeline of $Encoder_{eb}$ which converts a feature sequence into an embedding. Class-level learning is employed to train $Encoder_{eb}$ by using our proposed proxy-based PAM-Loss. One anchor sample and some augmented samples are generated from an original audio by using $Encoder_{fp}$. Three measures including Proxy-anchor Alignment (PA), Overlap-aware Margin (OM), and Proxy Initialization (PI) are integrated in PAM-Loss (see Section 3.3.2 for details).**

, where $m > 0$ represents an additive angular margin, and $s$ is the scaling parameter.

ArcFace-Loss hasn't found effective application in MIR, especially in audio identification. Triplet and N-pairs losses play more important roles in these tasks. It's evident that loss functions without proxies offer a more straightforward approach for retrieval tasks. Furthermore, **ArcFace exhibits some limitations when applied to variable-length audio embeddings** as follows:

(1) Firstly, ArcFace's primary objective is to drive the embeddings of various augmented fragment samples from the same music track to converge towards their proxies. However, in practical audio identification, the central goal is to accurately match a distorted audio snippet to a complete high-quality audio in databases. **The two objectives are not entirely equivalent, as ArcFace lacks a mechanism to align the proxy with the embeddings of such high-quality audio**.

(2) Secondly, ArcFace incorporates a fixed angular margin into its loss function. However, it becomes **unreasonable to persist with this fixed angular margin** when dealing with our task, where the degree of content overlap between the augmented sample and the anchor sample can vary.

(3) Finally, but **most importantly**, a significant challenge arises during the training process—ensuring the convergence of embeddings from different augmented samples towards their respective proxies. The diverse content of augmented samples within the same class may lead to weak relationships between them, because each augmented sample may only contain a fraction of the information present in the corresponding anchor, resulting in a lack of complete class knowledge. Consequently, it is unreasonable to require that these embeddings closely resemble the proxy **when the proxy itself lacks complete class knowledge**. This requirement for full class knowledge in the proxy is crucial for effective

training, distinguishing it significantly from other scenario, such as face recognition, where each training sample inherently encapsulates the entire class information. Experiments in Section 4.4.2 demonstrate this limitation apparently.

Notably, most MIR tasks heavily rely on pair-wise instead of class-lever learning. However, this limitation does not stem from the inherent capabilities of class-level learning itself but rather from the fact that commonly used class-level losses are not well-suited for MIR tasks.

*3.3.2 Proposed PAM-Loss.* We propose PAM-Loss that extends ArcFace-Loss by incorporating *proxy-anchor consistency*, *overlap-aware margin*, and *proxy initialization* to effectively address the three limitations outlined above.

Assuming there are $N$ samples in a training batch, and $M$ samples of them are original anchor samples. These $N$ samples are denoted as $\{Y_i, l_i, \gamma\}_{i=1}^N$, where $Y_i$ and $l_i$ share the same definition as in Section 3.3.1, and $\gamma$ indicates that the training sample is an anchor sample ($\gamma = 1$) or an augmented sample ($\gamma = 0$). The objective of PAM-Loss is expressed as

$$L_{pam} = \lambda_{align} \cdot \frac{1}{M} \sum_{i=1}^{M} L_{align_i} + \frac{1}{N} \sum_{i=1}^{N} L_{oam_i} \quad (2)$$

, where

$$L_{align_i} = \gamma \cdot (1 - \cos \theta_{l_i}) \quad (3)$$

$$L_{oam_i} = -\log \frac{e^{s \cos(\theta_{l_i} + m(\delta_i))}}{e^{s \cos(\theta_{l_i} + m(\delta_i))} + \sum_{j \neq l_i} e^{s \cos \theta_j}} \quad (4)$$

. $L_{align_i}$ is the proxy-anchor alignment loss and $L_{oam_i}$ is the overlap-aware margin loss of the $i$-th sample. The hyper-parameters $\lambda_{align}$ is used to trade-off between two losses. $\delta_i$ represents the overlapping duration ratio to $Dur_{max}$. Following measures are integrated in PAM-Loss to overcome the limitations of ArcFace-Loss:

**Proxy-anchor Alignment (PA)**. $L_{align_i}$ represents a straightforward angular cosine embedding loss function. The primary objective of this loss is to maintain consistency between the embedding of the **anchor** sample and its associated proxy. Given that the training dataset predominantly consists of high-quality samples, this loss function establishes a mechanism for aligning the proxy with the embedding of the complete high-quality audio. This alignment ensures that the training objective of the PAM-Loss aligns perfectly with the application goal of audio identification.

**Overlap-aware Margin (OM)**. $L_{dam_i}$ is derived from ArcFace-Loss with a dynamic overlap-aware margin $m(\delta_i)$. We define function $m(\delta_i)$ as a linear function as

$$m(\delta_i) = \delta_i \cdot (m_u - m_l) + m_l \tag{5}$$

, where $m_u$ and $m_l$ are the upper and lower of the angular margin range. This design follows a natural intuition: the embeddings of long samples should be closer to class proxy $P_{l_i}$ via $m(\delta_i)$. $L_{dam}$ **simultaneously further strengthens the consistency** between class proxies and their corresponding original samples (their duration ratio $\delta_i = 1.0$).

**Proxy Initialization (PI)**. As mentioned above, applying a proxy-based loss in audio identification requires that the proxy contains complete class knowledge in the training process. Otherwise, achieving convergence becomes challenging. Assuming there is an *original* music $X$ in the training dataset, its class label is $l$. We can initially convert $X$ into a feature sequence $\{z_t\}_{t=t_1}^{t_n}$ in Fingerprinting Stage. We have empirically found that the mean of these features, $z_{mean} = \frac{1}{n} \sum_{t=t_0}^{t_n} z_t$, is an excellent initial value for the corresponding class proxy $P$, because $z_{mean}$ effectively encapsulates knowledge of the class, as demonstrated in the experimental section. **This initialization significantly improve the performance** of $Encoder_{eb}$, and also serves as a compelling demonstration of **the necessity for a two-stage embedding** process, as $z_{mean}$ is a result of Fingerprinting Stage.

## 3.4 Training Strategy

A proxy should possess full knowledge of its corresponding anchor sample. *Proxy Initialization* greatly alleviates this challenge. Furthermore, we dynamically adjust the distribution of $\delta$ to further enhance the consistency between proxies and their anchor samples. $\delta$ serves as a measure of the proportion of full knowledge contained in the augmented samples. As we initiate the training process of $Encoder_{eb}$, our objective is to construct proxies that contain full knowledge, so we predominantly utilize samples with larger $\delta$. As the training processes, we gradually increase more samples with smaller $\delta$ that are considered challenging samples with lower knowledge content as hard-sample mining. Hence, we employ a function $\delta_i = \alpha^\beta$, where $\alpha \sim \text{Uniform}[0, 1]$ and $\beta \in [0.5, 3.0]$ is used to adjust the distribution, to randomly generate the overlap duration ratio $\delta_i$ for the augmented samples.

## 4 Experiment

## 4.1 Experimental Setting

*4.1.1 Dataset.* The evaluation for comparison was conducted on FMA[7] for training and testing. More specifically, four data sets isolated from each other are used:

(1) *Training-FP-DB*: a subset of $fma\_medium$ consisting of $30s$ audio clips from a total of 10K songs.
(2) *Traning-EB-DB*: a subset consisting of 30s audio clips from a total of 100K songs.
(3) *Test-Dummy-DB*: a subset of the $fma\_full$ consisting of about $278s$ audio clips from a total of 100K songs.
(4) *Test-Query-DB*: a subset of $fma\_medium$ consisting of 500 audio clips of 30s each. There is no overlap between Test-Query-DB and two training datasets.

*4.1.2 Implement Details.* AMG-Embedding was implemented in Pytorch framework. As the SOTA open source fingerprinting solution, the encoder of *neural fingerprinting*[4] is adopted as $Encoder_{fp}$. Initially, we trained $Encoder_{fp}$ using the source codes, settings and *Training-FP-DB* provided by the authors of *neural fingerprinting*. As standard Transformer Encoders, $Encoder_{eb}$ models share a hidden feature dimension of 256, 8 attention heads, and consist of 10 encoder layers. These models were trained using the Adam optimizer on the dataset named *Traning-EB-DB*. The initial learning rate of 1e-3 was reduced to 1e-4 using cosine decay over 50 epochs. A training batch has 640 samples, in which augmented samples and original anchor samples are mixed in a 9:1 ratio to balance $L_{oam}$ and $L_{align}$. We set $m_u = 0.4$, $m_l = 0.2$, scaling $s = 32$, and $\lambda_{align} = 10.0$ in all experiments. All encoders were trained on a server with 8-TitanX GPUs, and the retrieval results were obtained using a standard PC with an Intel Core i7-10700K CPU, 128GB RAM, NVIDIA GeForce RTX 3060 GPU with 12GB VRAM, and SSD hard disks.

**Querying method**. The retrieval database consists of audio songs from *Test-Dummy-DB* and *Test-Query-DB*, each assigned a unique ID and converted into embeddings. To retrieve, a random clip from *Test-Query-DB* is converted into embeddings. For each query embedding, top $K$ similar embeddings are retrieved from the database. Cumulative similarity scores are calculated for embeddings sharing the same song ID, and the song IDs are ranked based on these scores.

**Evaluation metric**. We use retrieval *accuracy*, retrieval *duration*, and indexing *feature size* together to evaluate the performance. *Top-1 hit rate* is employed to measure the accuracy. The metric of *Top-1 hit rate*(%) is defined as

$$Acc = 100 \times \frac{(n\ of\ hits\ @Top1)}{(n\ of\ hits\ @Top1) + (n\ of\ miss\ @Top1)} . \tag{6}$$

**Table 1: Comparisons with some classic methods.** $Size_{bd}$ represents the fingerprint or embedding count for building the retrieval database. $DUR_{qr}$ represent the average duration of querying a $30s$ audio.

| Method | Top-1 Acc.(%) by Query Lengths | | | | | $Size_{bd}$ | $DUR_{qr}$ |
|---|---|---|---|---|---|---|---|
| | 2s | 3s | 5s | 10s | 30s | | |
| Shazam[41] | 9.4 | 21.6 | 40.2 | 61.3 | 79.2 | 1.9 G | > 18 s |
| Dejavu[12] | 13.4 | 26.6 | 51.6 | 78.6 | 95.2 | 1.4 G | ≈ 13 s |
| NP[16] | 67.2 | 78.3 | 86.5 | 95.5 | 98.8 | 53.8 M | 8.71 s |
| NFS[4] | **89.7** | **94.2** | **97.6** | **99.3** | **100.0** | 53.8 M | 8.71 s |
| Ours | 87.7 | 93.2 | 96.5 | 98.4 | 99.4 | **5.93** M | **0.12** s |

**Table 2: Effect of AMG-Embedding compared with *neural fingerprinting* system(NFS)[4]. $Size_{bd}$ represents the embedding count for building the retrieval database. $DUR1_{bd}$ and $DUR2_{bd}$ are the duration of building retrieval database using Flat and IVF-PQ index in Faiss[22], and the duration associated with AMG-Embedding include inference time of $Encoder_{eb}$. $DUR1_{qr}$ and $DUR2_{qr}$ represent the average duration of querying a $30s$ audio using Flat and IVF-PQ index, and the duration include the inference time of $Encoder_{fp}$ for all methods and additionally $Encoder_{eb}$ for AMG-Embedding.**

| Method | $d\&f$ | $Dur_{max}$ | Top-1 Acc.(%) by Query Lengths | | | | | $Size_{bd}$ | $DUR1_{bd}$ | $DUR2_{bd}$ | $DUR1_{qr}$ | $DUR2_{qr}$ |
|--------|--------|-------------|------|------|------|------|------|-------------|-------------|-------------|-------------|-------------|
| | | | 2s | 3s | 5s | 10s | 30s | | | | | |
| NFS[4] | 128 | - | 89.7 | 94.2 | 97.6 | 99.3 | 100.0 | 53.8 M | 214.8 s | 305.9 s | 112.16 s | 8.71 s |
| Ours | 128 | 5s | 87.7 | 93.2 | 96.5 | 98.4 | 99.4 | 5.93 M | 421.56 s | 433.06 s | 1.04 s | 0.12 s |
| Ours | 128 | 10s | 77.3 | 87.6 | 96.0 | 99.2 | 99.8 | 2.79 M | 386.21 s | 393.57 s | 0.32 s | 0.06 s |
| Ours | 128 | 15s | 71.1 | 85.1 | 95.4 | 98.6 | 99.8 | 1.81 M | 357.79 s | 361.55 s | 0.20 s | 0.03 s |
| Ours | 128 | 30s | 63.5 | 79.2 | 90.3 | 98.0 | 99.6 | 0.87 M | 329.01 s | 334.49 s | 0.11 s | 0.03 s |
| NFS[4] | 256 | - | 94.5 | 97.5 | 99.3 | 99.9 | 100.0 | 53.8 M | 322.92 s | 720.75 s | 253.24 s | 8.94 s |
| Ours | 256 | 5s | 90.1 | 93.9 | 97.1 | 99.2 | 99.8 | 5.93 M | 521.04 s | 538.20 s | 2.57 s | 0.14 s |

## 4.2 Comparisons

We firstly compare our work with classic systems, including two handcrafted feature systems (our imeplemented *Shazam*[41] and a popular Shazam-style open-source *Dejavu*[12]), and two deeplearning systems (*Now-playing(NP)*[16], and *Neural fingerprinting system(NFS)*[4]). The fingerprints of *NP* and *NFS* were built with fixed-duration segments of $1s$ and $0.5s$ overlap. Moreover, the encoder of *NFS* is also employed as $Encoder_{fp}$ in AMG-Embedding. We created 21 query segments for each audio file in the *Test-Query-DB*, which includes the original $30s$ audio and 5 randomly cropped segments for each duration: $2s$, $3s$, $5s$, and $10s$. Therefore, this results in a total of 10,500 test query segments. Each query is synthesized through the quality augmentation pipeline used in *neural fingerprinting*[4]. All dimensions of the embedding are set to 128 for Now-playing, NFS and AMG-Embedding. In AMG-Embedding, we trained the $Encoder_{eb}$ with $Dur_{max} = 5s$. In *NP*, *NFS*, and AMG-Embedding, GPUs were employed during encoder inference, whereas the retrieval was performed using the CPU **without parallel processing (querying one embedding at a time)**. The Top1 accuracy results for AMG-Embedding are obtained using the Flat index in Faiss[22], while for *NFS*, due to its computationally intensive nature (see Section 4.3), the results are derived from the IVF-PQ index with a product quantizer (PQ) with the configuration: 256 centroids, each with a code size of 64 and 8 bits per index.

The results are listed in Table 1. It is evident that hand-crafted feature methods exhibit significantly lower performance in retrieval accuracy, retrieval duration, and feature size. The results also indicate that *NFS* achieves the highest retrieval accuracy. However, it does not exhibit advantages in retrieval duration and indexing feature size. Given that our proposed AMG-Embedding builds upon the foundation of *NFS*, **the primary objective of experiments** is to demonstrate that the AMG-Embedding can effectively preserve the accuracy of the underlying *NFS* while achieving a substantial reduction in the embedding size and the retrieval times. Therefore, we conducted the following experiments to further demonstrate the effectiveness of AMG-Embedding.

## 4.3 Effect of AMG-Embedding

To comprehensively demonstrate the effectiveness of our method, we trained several $Encoder_{eb}$ with different $Dur_{max} = \{5s, 10s, 15s, 30s\}$, and different dimension $d = f = \{128, 256\}$ for comparisons. As shown in Table 2, *NFS* achieves the higher accuracy, and the accuracy of AMG-Embedding is influenced by the $Dur_{max}$ apparently, with shorter durations yielding higher accuracy. Additionally, as the retrieval segment becomes shorter, the accuracy gap with *NFS* becomes more noticeable. However, once the queried segment exceeds 5 seconds, the accuracy gap between AMG-Embedding and *NFS* becomes minimal. Furthermore, a larger dimension of embeddings is preferred for higher retrieval accuracies. The final embedding quantity generated by AMG-Embedding is significantly reduced. Beyond the storage advantage, this reduction brings efficiency benefits to retrieval process.

As for querying duration and indexing feature size, we initially employed the Flat index in Faiss for comparisons. Table2 illustrates that the time required to build retrieval databases (referred to as $DUR1_{bd}$) is roughly comparable between AMG-Embedding and *NFS*. Of particular note is the negligible duration difference between AMG-Embedding and *NFS* in building retrieval databases, especially when compared to the nearly 5-hour consumption on fingerprinting computation shared by *NFS* and AMG-Embedding. However, *NFS* necessitates more query times due to the larger number of features it generates. Additionally, the Faiss feature index created by *NFS* is notably large, leading to a significant increase in single-query duration. Consequently, the time required for *NFS* to complete retrievals for a long audio sample is significantly greater than that of AMG-Embedding. When querying a $30s$ audio, *NFS* generates 59 embeddings, while AMG-Embedding ($Dur_{max} = 10s$) generates only 3 embeddings. Thus, their querying durations were $112s$ and $0.32s$, respectively.

Secondly, we adopted the IVF index to reduce the querying duration. While this technique enhances efficiency ($DUR2_{qr}$) significantly, it is essential to acknowledge the costs: a potential slight decrease in retrieval accuracy and an extra time-consuming task of computing centroids (see $DUR2_{bd}$). Considering the $30s$ audio, the querying durations of *NFS* and AMG-Embedding were $8.71s$ and

**Table 3: Top-1 retrieval accuracy and parameter size for different backbones of $Encoder_{eb}$.**

| Backbone | Top-1 Acc.(%) by Query Lengths | | | | | #Params |
|---|---|---|---|---|---|---|
| | 2s | 3s | 5s | 6s | 10s | |
| ResNet-18 | 68.7 | 81.6 | 94.8 | 96.4 | 98.6 | 12.0 M |
| ResNet-50 | 68.5 | 80.4 | 95.2 | 96.7 | 98.6 | 26.6 M |
| Bi-LSTM | 70.0 | 82.6 | 95.4 | 96.8 | 98.6 | 11.9 M |
| DenseNet121 | 70.8 | 82.9 | 95.7 | 96.4 | 98.6 | 8.1 M |
| EfficientNet-B3 | 73.6 | 85.7 | **96.2** | 97.0 | 98.9 | 12.4 M |
| Transformer | **77.3** | **87.6** | 96.0 | **97.2** | **99.2** | **4.4** M |

0.06$s$, respectively. It is evident that additional acceleration techniques, such as multi-thread computing, GPU acceleration, better retrieval strategies, etc., are still necessary for querying lengthy audio using *NFS*.

In a summary, AMG-Embedding introduces additional computations and incurs a slight decrease in retrieval accuracy, but the reduction in embedding size results in a **dramatic advantage in retrieval efficiency and storage consumption**.

## 4.4 Ablation Study of Embedding Stage

In this section, we present a series of ablation experiments designed to highlight the effectiveness of the employed techniques. For the sake of universality, we set $Dur_{max} = 10s$ and $f = d = 128$, ensuring a standardized comparison. We also generated 2500 query segments for each query length in these experiments.

*4.4.1 Effect of Transformer Encoder.* AMG-Embedding employs a standard transformer encoder as $Encoder_{eb}$. We also trained CNN-based ResNet-18 and ResNet-50[20], DenseNet121[21], EfficientNet-B3[39] and a standard sequence-to-sequence Bi-directional 8-layer (256 dimension) LSTM[32] encoders for comparisons. Based on the results in Table 3, it is evident that, even with fewer parameters, Transformer-based encoders exhibit notable performance advantages compared to both CNN and LSTM encoders.

Furthermore, our $Encoder_{eb}$ can be regarded as a feature compression or dimensionality reduction method. Therefore, we also attempted to use machine learning dimensionality reduction techniques, including PCA, T-SNE, etc., to compress features. Because the input is variable-length features, padding is needed before applying the above methods. Nevertheless, our experimental results indicate that these methods cannot effectively represent the features, with accuracy scores consistently at 0 for variable duration retrieval.

*4.4.2 Effect of PAM-Loss.* Our proposed metric loss function, PAM-Loss, plays a crucial role in AMG-Embedding. To assess the impact of PAM-Loss, we trained $Encoder_{eb}$ with various loss functions, including Triplet-Loss, InfoNCE (N-Pairs/NT-Xent)-Loss, ArcFace-Loss, and PAM-Loss. The accuracy results are listed in Table 4. Among these losses, Triplet-Loss and InfoNCE-Loss are widely employed in MIR-related tasks for pair-wise learning. According to the results, we can see that these two losses can be utilized in our task, and InfoNCE-Loss outperforms Triplet-Loss significantly.

**Table 4: Comparison of retrieval accuracy for $Encoder_{eb}$ across different loss functions and ablation investigation of PI, OM, and PA enhancements. Note that $Encoder_{eb}$ trained with Arc-Face failed to converge.**

| loss | PI | OM | PA | Top-1 Acc.(%) by Query Lengths | | | | |
|---|---|---|---|---|---|---|---|---|
| | | | | 2s | 3s | 5s | 6s | 10s |
| Triplet[47] | - | - | - | 14.4 | 32.7 | 64.6 | 74.3 | 87.2 |
| InfoNCE[29] | - | - | - | 49.5 | 66.4 | 89.4 | 92.7 | 96.6 |
| Arcface[8] | ✗ | ✗ | ✗ | 0.0 | 0.0 | 0.0 | 0.0 | 0.0 |
| PAM | ✓ | ✗ | ✗ | 75.0 | 86.4 | 96.0 | 96.5 | 98.8 |
| | ✓ | ✓ | ✗ | 76.6 | 86.7 | **96.2** | 96.9 | 99.0 |
| | ✓ | ✓ | ✓ | **77.3** | **87.6** | 96.0 | **97.2** | **99.2** |

As discussed above, though ArcFace-Loss has achieved considerable success in numerous tasks, it is not suitable in variable length audio embedding. It is proved by results in Table 4. In fact, **the encoder trained with ArcFace-Loss failed to converge to a level suitable for practical applications**. PAM-Loss addresses the limitations of ArcFace-Loss in handling the variable-length embedding problem by incorporating three measures: *Proxy-anchor Alignment (PA), Overlap-aware Margin (OM)*, and *Proxy Initialization (PI)*. We gradually integrated them into ArcFace-Loss to demonstrate their effectiveness. According to the results in Table 4, it is obviously that each measure can boost the performance apparently, and *PI* plays the most important role in PAM-Loss. The results also demonstrate the effectiveness of *PA* and *OM*.

## 4.5 Limitations of AMG-Embedding

The primary goal of AMG-Embedding is to address challenges related to resource consumption and retrieval efficiency in large feature database generated by fingerprinting methods. However, AMG-Embedding, by design, does not excel in providing precise matching positions as fingerprinting. Retrieval accuracy of AMG-Embedding is also not expected to surpass the based fingerprinting method, especially for very short retrieval segments. Additionally, the Embedding Stage introduces additional computational overhead, but this overhead can be effectively mitigated by retrieval accelerations. As a result, AMG-Embedding is more suitable for scenarios involving large retrieval database where storage limitations and retrieval efficiency are critical considerations.

## 5 Conclusion

In this paper, we introduce AMG-Embedding, a novel approach to audio identification. Our main aim is to represent audio with much shorter embedding sequences compared to fingerprinting, while still supporting high-specific fragment-level retrieval. AMG-Embedding achieves this through a two-stage variable-length embedding using a Transformer encoder trained with PAM-Loss. PAM-Loss is also a valuable exploration for applying class-level learning in audio identification, showing potential for broader use in other MIR tasks. AMG-Embedding achieves accuracy levels comparable to fingerprinting solutions, yet it provides a significant advantage in retrieval efficiency and storage consumption.

## Acknowledgments

This work was supported in part by the National Natural Science Foundation of China under Grant No.62271034.

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
