# OpenReview forum: "AMG-Embedding: a Self-Supervised Embedding Approach for Audio Identification"
_acmmm.org/ACMMM/2024/Conference — MM2024 Poster_

### Official Review · Reviewer_6yT7 · 2024-05-18

**Rating:** 5
**Confidence:** 2

**Summary:**

This paper proposed a new self-supervised embedding method for audio identification named AMG-embedding. The AMG-embedding can represent the audio in a variable-duration manner, and improve the granularity and specificity.

**Strengths:**

1. This paper is very well written and easy to follow. The focuses are highlighted. The structure is clear and reasonable. The figures and tables are informative and easy to understand.

2. This paper proposed two-stage training where the second embedding stage could compress the fixed-duration audio features to smaller embeddings.

3. This paper proposed PAM-loss to replace ArcFace loss to improve variable-length embeddings.

4. The experiment results show that the proposed AMG-embedding significantly reduces the size of the retrieval database and the query duration.

**Limitations:**

1. The embedding stage is based on the fingerprinting stage (i.e., the NFS in Table 1). So why is the AMG-embedding slightly less accurate than NFS in query?

2. The duration for building a retrieval database of AMG-embedding is quite longer than NFS (in Table 2). Is it because the AMG-embedding the combination of NFS and embedding stage?

**Suitability:**

3

---

### Official Review · Reviewer_KrHc · 2024-05-20

**Rating:** 4
**Confidence:** 4

**Summary:**

This paper present AMG-Embedding, which is built on a traditional fingerprinting encoder and aims to represent variable-duration nonoverlapped segments as embeddings through a two-stage embedding. AMG-Embedding reduces the number of generated embeddings while achieving high-specific fragment-level audio identification.
Experimental results demonstrate that AMG-Embedding achieves retrieval accuracy comparable to the based fingerprinting approach while consuming less than 1/10𝑡ℎ of its storage and retrieval time.

**Strengths:**

1.	In light of the inefficiency of previous audio identification approaches, AMG-Embedding effectively reduces the size of feature sequences. The current fingerprinting approach can be seamlessly integrated into AMG-Embedding.
2.	Based on ArcFace loss, this paper introduces the design of PAM-loss. It thoroughly explains why ArcFace is ineffective for MIR tasks and how PAM-loss was designed. The ablation study validates the effectiveness of the PAM-loss, specifically highlighting the contributions of the PI, OM, and PA modules in PAM.
3.	The experiments concerning AMG-embedding are very comprehensive, particularly in terms of size and duration, which allow readers to clearly understand the efficiency of AMG-Embedding.

**Limitations:**

1.	Table 1 compares the baseline methods, which do not include the latest approaches. The focus is on comparing with NFS, and although the size is reduced to 1/10, the performance does not surpass NFS. NFS is a neural fingerprinting work published in 2021, and it would be valuable to include comparisons with the latest fingerprinting methods.
2.	In Table 4, I believe that PAM1, specifically PI, achieved the most significant performance improvement within the PAM loss. My question is why a margin of 0.3 was used when only PI was applied, given that the default margin in the original ArcFace paper is 0.5. This margin controls the clustering degree of class angles, and I suppose that an appropriate margin could potentially allow PI to outperform PI+OM. On the other hand, it would be insightful to explore whether PAM loss can form an effective feature space. Using T-SNE visualization might better illustrate the progression from the original ArcFace to PI, PI+OM, and finally to PI+OM+PA.

**Suitability:**

3

---

### Official Review · Reviewer_Ysex · 2024-05-24

**Rating:** 3
**Confidence:** 2

**Summary:**

The paper presents AMG-Embedding, a self-supervised embedding approach designed for audio identification. The method aims to improve the efficiency and scalability of audio retrieval systems by reducing the number of embeddings needed to represent audio tracks. Traditional fingerprinting methods require numerous short, fixed-duration, and overlapping segments, leading to large feature databases and high computational costs. AMG-Embedding addresses these issues by representing variable-duration non-overlapping segments through a two-stage embedding process. The approach integrates a fingerprinting encoder and a Transformer-based encoder trained with a novel loss function, Proxy-anchor Aligned Margin Loss (PAM-Loss). Experimental results demonstrate that AMG-Embedding achieves comparable retrieval accuracy to traditional methods while significantly reducing storage and retrieval time.

**Strengths:**

- The two-stage embedding process combining traditional fingerprinting with a Transformer-based encoder is well-conceived.
- The method significantly reduces the number of embeddings for audio identification.
- The reduced computational and storage requirements make AMG-Embedding suitable for large-scale audio databases.
- The experiments demonstrate that the proposed method is competitive concerning retrieval accuracy while offering efficiency improvements.
- The approach can integrate any state-of-the-art fingerprinting method.

**Limitations:**

- The wording uses 'audio' a lot while focusing on 'music'.
- Given that the first stage already exists in the literature, the overall contribution is rather shallow.
- The comparison is very limited. One could use traditional fingerprinting methods with less overlapped (or non-overlapped) sections to equal with AMG-Embedding in terms of effiency.
- MERT is not considered, which currently represents the state of the art for music audio embedding.
- In general, the evaluation does not showcase the impact of the proposed method in scientific or practical terms.

**Suitability:**

2

---

### Meta-Review · Area_Chair_JRbQ · 2024-07-11

**Recommendation:** Accept (Poster)
**Confidence:** 4

**Metareview:**

**Conclusion: Accept as a Poster Paper**

After thoroughly evaluating the three independent reviews, I recommend accepting this submission as a Poster Paper for ACM MM 2024. The submission is a significant contribution to the field of audio retrieval systems, addressing critical issues of efficiency and scalability. Although some limitations are noted, the strengths and innovative aspects of the approach justify its acceptance.

**Strengths:**

1. **Innovative Two-Stage Embedding Process**: The proposed method integrates traditional fingerprinting with a Transformer-based encoder, significantly reducing the number of embeddings needed for audio identification. This innovation effectively addresses the inefficiency of previous methods.
2. **Efficiency Improvements**: The method substantially decreases computational and storage requirements, making it suitable for large-scale audio databases. Experimental results show that AMG-Embedding achieves comparable retrieval accuracy while reducing storage and retrieval time to less than one-tenth of the traditional methods.
3. **Comprehensive and Clear Presentation**: The paper is well-written, with a clear and logical structure. The figures and tables are informative and easy to understand, providing a thorough explanation of the PAM-loss and its components.

**Weaknesses:**

1. **Limited Comparison with Latest Approaches**: The paper primarily compares AMG-Embedding with an older method (NFS) and lacks comparisons with the latest fingerprinting techniques, such as MERT, which represents the state-of-the-art in music audio embedding.
2. **Accuracy Trade-offs**: While AMG-Embedding significantly reduces database size and query duration, it slightly underperforms in terms of accuracy compared to the NFS method. This trade-off needs further investigation and justification.
3. **Database Construction Duration**: The duration for building a retrieval database is longer for AMG-Embedding than for NFS, likely due to the two-stage embedding process. This aspect requires further optimization to enhance the method's practical applicability.

In summary, the reviewers collectively acknowledge the innovative contributions and practical significance of AMG-Embedding in improving audio retrieval systems. The strengths in methodology, efficiency, and comprehensive presentation outweigh the noted limitations. Given the overall positive assessment and the potential impact on the multimedia community, this paper is recommended for acceptance as a Poster Paper at ACM MM 2024.